# Effect of Head-Up/-Down Tilt on ECG Segments and Myocardial Temporal Dispersion in Healthy Subjects

**DOI:** 10.3390/biology12070960

**Published:** 2023-07-05

**Authors:** Gianfranco Piccirillo, Federica Moscucci, Ilaria Di Diego, Martina Mezzadri, Cristina Caltabiano, Myriam Carnovale, Andrea Corrao, Ilaria Lospinuso, Sara Stefano, Claudia Scinicariello, Marco Giuffrè, Valerio De Santis, Susanna Sciomer, Pietro Rossi, Emiliano Fiori, Damiano Magrì

**Affiliations:** 1Department of Clinical and Internal Medicine, Anesthesiology and Cardiovascular Sciences, Policlinico Umberto I, “Sapienza” University of Rome, Viale del Policlinico, 00161 Rome, Italy; 2Department of Internal Medicine and Medical Specialties, Policlinico Umberto I, Viale del Policlinico, 00161 Rome, Italy; 3Arrhythmology Unit, Fatebenefratelli Hospital Isola Tiberina-Gemelli Isola, 00186 Rome, Italy; 4Department of Clinical and Molecular Medicine, Sant’Andrea Hospital, “Sapienza” University of Rome, 00198 Rome, Italy

**Keywords:** head-up tilt test, autonomic nervous system, heart rate variability, T wave amplitude

## Abstract

**Simple Summary:**

The autonomic nervous system (ANS) modulates the oscillation of electrocardiogram segments and their intervals so much that its balance could be investigated throughout the heart rate variability analysis. It is noteworthy that the ANS is able to modulate the myocardial repolarization phase too. Head-up/-down tilt test modifies acutely the ANS balance by means of a deactivation of the cardiopulmonary reflexes. The present study examines the influence of head-up/-down tilt on a number of ECG segments, assuming that the cardiopulmonary postural reflexes modulate most of the ECG interval oscillations in terms of their lengths and short-period temporal dispersion. The T wave amplitude diminished during head-up tilt and significantly correlated with the left ventricular end-systolic volume.

**Abstract:**

The head-up/-down tilt test acutely modifies the autonomic nervous system balance throughout a deactivation of the cardiopulmonary reflexes. The present study examines the influence of head-up/-down tilt on a number of ECG segments. A total of 20 healthy subjects underwent a 5 min ECG and noninvasive hemodynamic bio-impedance recording, during free and controlled breathing, lying at (a) 0°; (b) −45°, tilting up at 45°, and tilting up at 90°. Heart rate variability power spectral analysis was obtained throughout some ECG intervals: P-P (P), P-Q (PQ), P_e_Q (from the end of P to Q wave), Q-R peak (QR intervals), Q-R-S (QRS), Q-T peak (QT_p_), Q-T end (QT_e_), ST_p_, ST_e_, T peak-T end (T_e_), and, eventually, the T_e_P segments (from the end of T to the next P waves). Results: In all study conditions, the Low Frequency/High Frequency_PP_ and LF_PP normalized units (nu)_ were significantly lower than the LF/HF_RR_ and LF_RRnu_, respectively. Conversely, the HF_PP_ and HF_PPnu_ were significantly higher in all study conditions. ST_e_, QT_p,_ and QT_e_ were significantly related to the PP and RR intervals, whereas the T wave amplitude was inversely related to the standard deviations of all the myocardial repolarization variables and to the left ventricular end-systolic volume (LVEDV). The T wave amplitude diminished during head-up tilt and significantly correlated with the LVEDV.

## 1. Introduction

The autonomic nervous system (ANS) modulates the oscillations of the heart rate, intra-atrial, atrio-ventricular, and intra-ventricular conductions as well as myocardial repolarization. Although sinus node activity is not properly identifiable on a surface electrocardiogram (ECG) since it precedes the P wave, it is measured roughly in terms of R-R intervals [1,2]. It is noteworthy that each ECG segment (P-Q, P-R, S-T, and T-P), as well as ECG intervals (RR, PP, P wave, Q-R-S, Q-T), slightly oscillate due to the ANS control, which modulates sinus node activity, affecting the ECG’s spectral components and coherences. Accordingly, in a recent study [3] on healthy and chronic heart failure subjects, a significant relationship has been described between both P-Q and PP coherence at rest and between the left ventricular ejection fraction (LVEF) and atrial volume. Similarly, in an animal experimental model, an inverse relationship has been found between left stellate ganglion nerve activity and the temporal dispersion of the segments P_e_-Q, i.e., from the end of P to the Q waves, whereas the same parameter was positively related to the vagal nerve activity [4]. Adding further support to the abovementioned close relationships, an increase in temporal dispersion of the P wave and P-Q segments during tilt test-induced asystole is clear [5]. Conversely, with respect to the link between the ANS and myocardial temporal dispersion, evidence in an animal experimental model has already been provided, specifically about the increase in Q-T/R-R coherence during high levels of left stellate ganglion activity [6] as well as about the direct relationship between the short-term variability of the T peak to T end interval (Te) and left stellate ganglion activity [7]. Furthermore, the close relationship between the circadian rhythm of the repolarization phase and the sympathovagal circadian cycle has been demonstrated [8]. Finally, passive orthostatism, as obtained by means of a tilt test, is equally able to elicit a sympathetic drive and to induce a myocardial temporal dispersion reduction, as well as high levels of sympathetic stress, as obtained by an exercise test, increasing the Q-T and Te standard deviation [9]. 

The aim of the present study, conducted in healthy subjects, was to accurately describe the physiological behavior of a number of ECG and noninvasive hemodynamic parameters in response to an ANS deactivation induced by different passive graded head-up/-down tests. Data acquired will be useful to program artificial intelligence and machine learning instruments in order to better stratify patients’ risk during stroke, myocardial infarction, acutely decompensated chronic heart failure, or atrial fibrillation. 

## 2. Materials and Methods

### 2.1. Study Protocol

For the study purpose, only healthy volunteers were enrolled in the present analysis. All participants underwent a short-term (5 min) single-lead ECG and noninvasive hemodynamic recordings (PhysioFlow; Manatec Biomedical, Poissy, France) lying supine over the tilt-table during free breathing (rest) and during controlled breathing (15 breaths/minutes) lying with the following tilt angles: (a) lying supine at 0° (0° L-d); (b) lying down at −45° (−45° T), tilting up at 45° (45° T), and tilting up 90° (90° T). The order of the study phases was randomly chosen [10] and each one was always preceded and followed by a rest session of 5 min. Hemodynamic recordings have been double-checked by echocardiographic tests and twelve leads surface ECG. The PhysioFlow system provided a self-correction to minimize artifacts and noises.

Patients were randomly assigned to different study phases, as shown in Figure 1. 

ECG signals were acquired and digitalized with a custom-designed card (National Instruments, Austin, TX, USA) at a sampling frequency of 500 Hz. Measurements used for the ECG segments and interval analysis were detected automatically by a classic adaptive first derivative/threshold algorithm and a template method [5,11] (Figure 2). 

A specifically designed and produced system for data acquisition, storage, and analysis was developed with the LabView program (National Instruments, Austin, TX, USA). An expert cardiologist (GP) checked the different ECG intervals and segments, automatically marked by the software and, when needed, manually corrected the mistakes [3,4,5,11,12]. 

The mean and standard deviation (_SD_) values of the following ECG intervals and segments were calculated: R-R intervals (RR); P-P intervals (PP), from the beginning of two consecutive P waves; P wave intervals (P), from the start to the end of a single P wave; P-Q intervals (PQ), from the start of P to the Q waves; P-Q segments (P_e_Q), from the end of P to Q waves; Q-R intervals (QR), from Q to the peak of R waves; Q-R-S intervals (QRS), from Q to S waves; Q-T peak intervals (QT_p_), from Q to the peak of T waves; Q-T end (QT_e_), from Q to end of T waves; ST peak segments (ST_p_), from S to the peak of T waves; ST end segments (ST_e_), from S to the end of T waves; T peak T end intervals (T_e_), from peak to end of T waves; T end P interval (T_e_P), from the end of T to start of P waves (Figure 3). 

Power spectral analyses with an autoregressive algorithm for all study variables have been performed to obtain the main three spectral components from PP and RR variability [7,13,14,15,16] very low-frequency power (VLF), between 0 and 0.04 hertz equivalents (Hz); low-frequency power (LF), between 0.04 and 0.15 (Hz); and high-frequency power (HF). The ratios between LF and HF (LF/HF) and the total power (TP) were also calculated, the latter being the total area under the spectra (i.e., the variance of the examined variable). Eventually, the absolute power in the LF and HF normalized units (nu) [7,13,14,15,16] and the spectral coherence between different variables were obtained [6,11,17], the latter expressing the mutual influence between two variables (ranging from 0 to 1).

Bioimpedance cardiography was used to evaluate short-term non-invasive hemodynamic variations during different study phases. Specifically, we recorded the stroke volume (SV), cardiac output (CO), systemic peripheral resistances (SPR), left ventricular end-diastolic volume (LVEDV), left ventricular end-systolic volume (LVESV), and left ventricular ejection fraction (LVEF) [18,19,20,21,22,23,24]. 

### 2.2. Statistical Analysis 

Data were reported as mean ± standard deviation or as median and interquartile range, respectively, for normal and skewed distribution data. We used the ANOVA for repeated measures to compare the studied variable (rest, 0° L-d, −45° T, 45° T, and 90° T) for the normally distributed variables and Friedmann and Wilcoxon’s tests for those non-normally distributed. Linear regression analysis was used to evaluate the influence between beat-to-beat PP or RR and all study variables (more than 40,000 points) and the influence of the amplitude of T on standard deviation repolarization of data. We also calculated the Pearson/Spearman correlation coefficients between noninvasive hemodynamic and ECG data, considering altogether the different study phases. 

Statistical analysis was performed using SPSS-PC+ (SPSS-PC+ Inc., Chicago, IL, USA) packages. All tests were two-sided. A *p* value less than or equal to 0.05 was considered as statistically significant.

## 3. Results

A total of 20 healthy subjects (mean age: 35 ± 14 years, 75% male) concluded the protocol. Table 1 shows in detail data from PP power spectral analysis measured in each of the five study phases and possible differences with respect to the data from RR power spectral analysis (Table 1). 

In particular, the LF/HF_PP_ and LF_PPnu_ were significantly lower than LF/HF_RR_ and LF_RRnu_, respectively, whereas the HF_PP_ and HF_PPnu_ were significantly higher than the same spectral components calculated on the RR (Table 1). All spectral components expressed in normalized power as well as the LF/HF ratio showed the most significant changes during the different study phases. It was noteworthy that all study phases, except for the “rest” one, were obtained during controlled breathing at 15 breaths per minute. This aspect explains why the central frequency at rest was significantly different from all other study phases. In all study conditions, the P-PP, PQ-PP, and T_e_P-PP coherences were significantly higher than the same coherences obtained with RR (numerical data not extensively reported), with the P-PP and PQ-PP coherences being significantly higher during the 90° T phase in comparison to the other study conditions.

Table 2 supplies a detailed list of all ECG intervals and segments data recorded during the five study phases. Obviously, the shortest RR and PP intervals were recorded during the 90° T phases, whereas the lowest were obtained at the −45° T phase. 

The P, PQ, and P_e_Q intervals were significantly shorter during the 90° T phase than in all other study conditions. Conversely, the QR and QRS intervals as well as P_SD_, P_e_Q_SD_, PQ_SD_, QR_SD_, and QRS_SD_ did not change significantly among the different study conditions. The lowest T wave amplitude was observed during the 90° T phase, whereas the highest one has been found during the 0° L-d phase. The shortest values for QT_p_, QT_e_, ST_p,_ and ST_e_ were obtained during the 90° T phase, while most of the myocardial repolarization variables were longer at −45° than rest, 0° L-d, and 45° T. Interestingly Te remained substantially unchanged under all the study phases. During the 90° T phase, most of the myocardial temporal dispersion repolarization variables (i.e., the standard deviation for all QT and ST segments analyzed) were significantly higher than at rest. During the 90° T phase, the value of the T_e_P segment was the shortest.

Table 3 shows the results from the multiple linear regression analysis between the PP and RR intervals and all the other ECG data. Besides the expected impact of the heart rate changes, the strongest relationships have been found between all RR or PP intervals and ST_e_ (PP or RR, r: 0.62), QT_p_ (PP, r: 0.64; RR, r: 0.65), QT_e_ (PP, r: 0.74; RR, r: 0.75), and T_e_P (PP, r: 0.97; RR, r: 0.96), (Table 3) (Figure 4).

The logarithm of the T wave amplitude was inversely related to the logarithm of standard deviations of all myocardial repolarization variables regardless of the study phase (logT wave amplitude-ST_pSD_, r: 0.24, *p* < 0.001; logT wave amplitude-ST_eSD_, r: 0.3, *p* < 0.001; logT wave amplitude-QT_pSD_, r: 0.37, *p* < 0.001; logT wave amplitude-QT_eSD_, r: 0.38, *p* < 0.001; logT wave amplitude-T_eSD_, r: 0.59, *p* < 0.001); (Figure 5). 

Regarding the noninvasive hemodynamic measurements, during the 90° T phase, the stroke volume significantly decreased compared to other study conditions (63 ± 22 mL versus rest: 84 ± 24 mL; 0° L-d: 82 ± 23 mL; −45° T: 81 ± 26 mL; 45° T: 73 ± 22 mL, *p* < 0.05 for all), with the cardiac output remaining almost unchanged (i.e., a physiological counterbalance between stroke volume and heart rate). Eventually, the left ventricular end-diastolic volume during the 90° T phase was significantly lower (114 ± 44 mL) than at rest (130 ± 35 mL, *p* < 0.05) and during 0° L-d (128 ± 34 mL, *p* < 0.05). A positive relationship was found between the left ventricular end-systolic volume and QR segments (r: −0.35, *p* < 0.001) and a negative one between the T wave amplitude and the left ventricular end-systolic volume considering all the study conditions together (r: −0.25, *p* < 0.05) (Figure 6).

## 4. Discussion

The present study aimed to describe in detail, in a physiological setting, the behavior of a large number of ECG parameters as well as of the main hemodynamic variables in response to a sympathovagal imbalance induced by different graded head-up/-down tests. Accordingly, the main findings might be summarized as follows: (a) PP or RR spectral components of variability do not completely overlap in their changes; (b) most of the short-period temporal dispersion variables are influenced by the sympathovagal imbalance induced by the head-up/down tilt maneuver; (c) heart rate changes do not impact identically on the different ECG variables; (d) passive standings induce a T wave amplitude flattening and an increase in the myocardial repolarization dispersion markers; and (e) the T wave amplitude is inversely related to the end-systolic volume obtained across all the different study phases.

### 4.1. PP and RR Variability and Coherences

Present data overlap with those obtained in previous studies dealing with HRV power spectral analysis during tilt. Specifically, during the maximal sympathetic drive (90° T phase), we demonstrated an increased LF_PP_nu, LF_RR_nu, LF/HF_PP_, and LF/HF_RR_ with a concomitant reduction in HF_PPnu_, HF_RRnu_, as well as PP and RR intervals shortening. These data suggest a sympathetic prevalence. Indeed, the normalized LF and HF power are well-known markers of sympathetic and vagal modulation of the sinus node activity, respectively [15,16,17,18,19]. Thus, our findings support the passive standing as a maneuver able to induce a significant sympathetic stimulation and parasympathetic inhibition and, contextually, they confirm the PP and RR power spectral components as useful noninvasive ANS balance markers. The abovementioned phenomenon was observed both at 90° and 45° tilt angles. Being at 45° T somewhat mitigated than at 90° T. This behavior suggests a gradualness of sympathetic activation and vagal inhibition in response to the tilt-table angle [25]. Conversely, during the −45° T phase, we registered a significant increase in PP and RR intervals as well as in HF_PPnu_ with a concomitant reduction in LF/HF_PP_ and LF/HF_RR_, this behavior likely indicating a vagal sinus node activity increase [26]. Of note, albeit the PP and RR variability (i.e., standard deviation) were equal, their normalized power of the spectral components and the corresponding values of LF/HF and HF were different. Particularly, the HF_PP_ was significantly higher than HF_RR_, and LF/HF_PP_ and LF_PPnu_ were significantly lower than the respective components obtained from RR spectral analysis. 

### 4.2. P Wave and Atrioventricular Conduction Variability

We observed an intra-atrial conduction increase during the 90° T phase (i.e., maximal sympathetic stimulation) and an opposite behavior during the rest and 0° L-d phases. The same behavior was observed for the PQ and P_e_Q segments, which were shortened significantly during the 90° T phase. Particularly, the P_e_Q segment, closely depending on atrioventricular conduction system function (AV node, His bundle, bundle branch, Purkinje fibers), diminished by about 15–20% in comparison to the rest and 0° L-d phases. Thus, our data suggest that the maximum effect on PQ segments (PQ = P + P_e_Q) of the sympathetic stress is limited to the P_e_Q (i.e., the atrioventricular node and His-Purkinje system), with the result quite expected given the wide sympathetic innervation reported at this level [27]. Eventually, the P, P_e_Q, and PQ segments were weakly influenced by heart rate, the latter exerting its influence predominantly on the myocardial repolarization phase [28]. Indeed, the ANS activity does not impact the short period P, P_e_Q, and PQ temporal dispersion, as expressed in terms of standard deviation, in the healthy subject, whereas it is possible that it happens in some settings of high risk for atrial fibrillation or supraventricular brady/tachyarrhythmias [7,13,14,15,16,29]

### 4.3. Intrinsicoid Deflection, Intraventricular Conduction, and Repolarization Variability

According to the present findings, both QR and QRS intervals, also called “R-peak time” or “intrinsicoid deflection” and “intraventricular conduction”, and their variability (in terms of standard deviation) seem not to be modulated by the ANS activity due to the fact that they remained almost unchanged during all the five study phases. Notwithstanding, the intrinsicoid deflection obtained by unipolar leads (V5–V6) has been suggested as a possible non-invasive electrical marker of sudden cardiac death [30,31]. In the actual study, we calculated the R-peak time just on the II lead and in a healthy setting and analyzed this variable only in terms of its possible oscillatory behavior within the P-QRS-T complex. Of note, the R-peak time was found to relate to the left ventricular end-systolic volume, suggesting that this parameter might be influenced by the myocardial structure rather than by the ANS itself [30]. 

Conversely, the myocardial repolarization length and its variability are known to be strongly influenced by both the ANS [6,7,8,9,11] and the heart rate [32]. Accordingly, the heart rate correction is widely accepted for the QT segment corrections, albeit it still remains controversial for T_e_, which seems less influenced by the RR segment [33,34]. Particularly, the shortest QT_p_, QT_e_, ST_p_, and ST_e_ were observed during the 90° T phase (i.e., maximal postural sympathetic stimulation) while they increased maximally during the −45° T phase (i.e., maximal postural vagal stimulation). It is noteworthy that just the T_e_ remained unchanged in all study conditions, reinforcing the leading idea about its poor dependency on the ANS balance. On the other hand, the myocardial temporal dispersion, expressed in terms of standard deviation values, was always significantly higher during sympathetic drive with the highest QT_pSD_, QT_eSD_, ST_pSD_, ST_eSD_, and T_eSD_ observed during the 90° T phase. Moreover, the T_eSD_ was also influenced by the vagal stimulus, being significantly increased during the −45° T phase. Notably, the same behavior was observed for the QT_eSD_ and ST_eSD_ but, given that QT_pSD_ and JT_pSD_ were not different, this finding seemed to be driven by the T_eSD_. Eventually, the T_e_P_SD_, which is the square root of the variance (or Total Power) of this variable, was equal to PP_SD_ and RR_SD,_ suggesting a de facto overlap between the RR or PP variability and the T_e_P variability. Accordingly, the coherence between RR or PP and T_e_P was optimal in all study conditions.

### 4.4. T Wave Amplitude

Recent studies reported that T wave amplitude might influence the measures of repolarization variability [11,35]. Specifically, the variability was found to be inversely related to the T wave amplitude and, in order to clarify this controversial issue [36], we also investigated and confirmed an inverse relationship between T wave amplitude and QT_pSD_, QT_eSD_, and T_eSD_. Furthermore, we found that the T wave template was flatter during the 45° T and 90° T phases with a T wave amplitude lower with respect to the other study phases. In such a context, we want to emphasize that T amplitude and end-systolic volume were inversely related, thus allowing us to speculate that this behavior might depend on a redistribution of the myocardial blood flow during the systole. 

### 4.5. Clinical Applications

The clinical application of these findings may be particularly interesting in the field of acute and chronic heart failure. Indeed, we know that during acutely decompensated chronic heart failure an important portion of the ANS is unbalanced with a hyper arousal of the sympathetic nervous system. The modulation of ANS activity and the influence that the patient’s position seems to have during recumbent or passive orthostatism on the ventricular filling pressures could guide the clinician in the diagnostic and therapeutic choices. Moreover, the use of these data in artificial intelligence and machine learning instruments will enhance the diagnostic power of ECG analysis, improving patients’ prognosis (sudden cardiac death, acutely decompensated chronic heart failure, etc.).

## 5. Conclusions

The present physiological study allows a better understanding of the close relationship between the surface ECG and the autonomic nervous system modulation. Indeed, adopting the short-term PP and RR variability power spectral analysis, it describes the ANS activity oscillations induced by postural deactivation of the cardiopulmonary reflexes during different passive graded head-up/-down tilt tests. Contextually, it supplied a detailed picture of the ECG intervals and hemodynamic parameter changes as well as their relationships in response to the abovementioned ANS oscillations.

## 6. Limitations

This study aims to evaluate the interactions between the autonomic nervous system, ventricular filling pressures, and electrocardiographic variations from a physiological point of view. This aspect obviously makes the clinical applicability limited. However, the study lays the foundations for a more in-depth understanding of the mechanisms that regulate the activity of the ANS, thus guaranteeing the possibility of future studies in pathological conditions (heart failure, acute myocardial infarction, acute pulmonary edema, etc.) to try new diagnostic and therapeutic approaches.

The small sample size requires future and larger studies in order to validate the data obtained.

## Figures and Tables

**Figure 1 biology-12-00960-f001:**
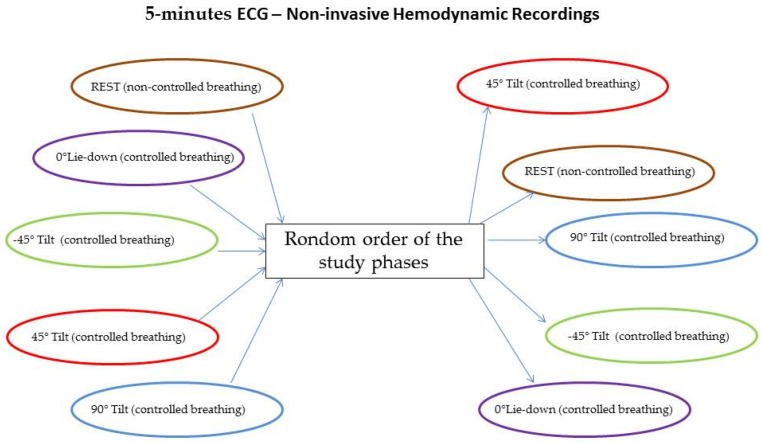
Random assignment to different study phases.

**Figure 2 biology-12-00960-f002:**
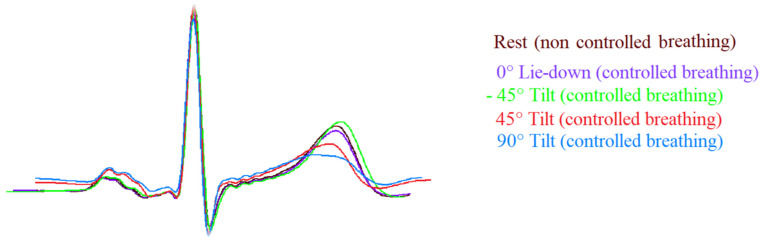
Template of ECG signals (II lead) during the different study phases. Note the T wave amplitude was flatter during passive positive standing with 45° (45° T, red) and 90° (90° T, blue) tilt table angle and it was sharper during head-down (−45° T, green) and rest (brown) in comparison to lay-down (0° L-d, purple).

**Figure 3 biology-12-00960-f003:**
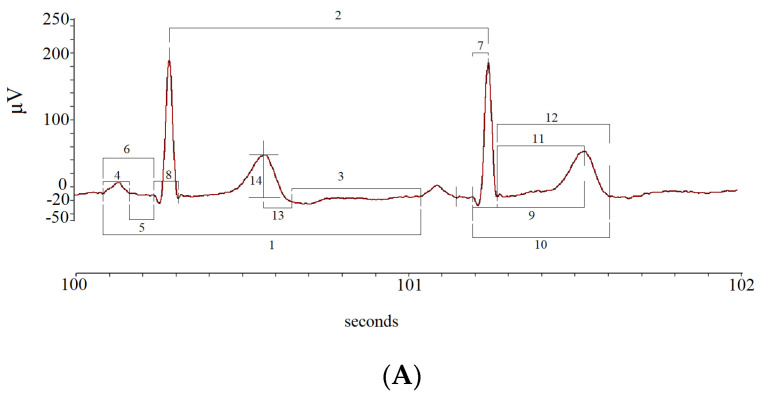
Segments and intervals obtained during the different study phases (**A**): rest (lay-down), controlled breathing (lay-down), head-down −45° tilt angle (−45° T), head-up 45° tilt angle (45° T), and head-up 90° tilt angle (90° T) (**B**).

**Figure 4 biology-12-00960-f004:**
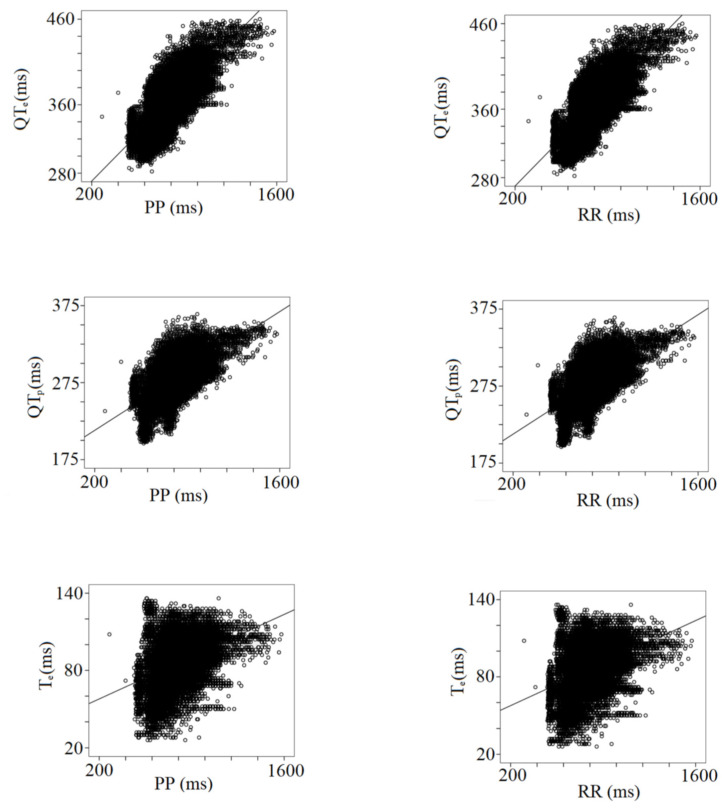
Correlation between PP or RR intervals with repolarization data, QTe (from q to the end of T wave), QTP (from q to the peak of T wave), and Te (from the peak to the end of T wave).

**Figure 5 biology-12-00960-f005:**
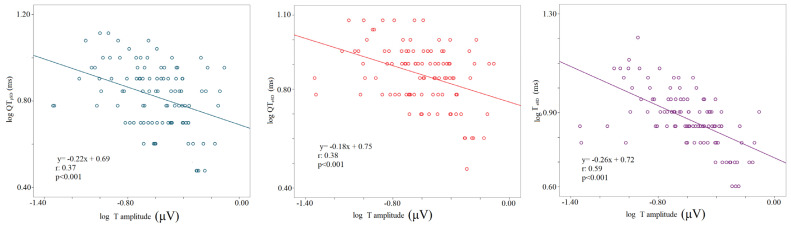
Correlation between T wave amplitude and repolarization temporal dispersion data, the logarithm of voltage (mV), the logarithm of the standard deviation of QTp (QTpSD) (**left** panel), QTe (QTeSD) (**middle** panel), and Te (from the peak to the end of T wave) (**right** panel).

**Figure 6 biology-12-00960-f006:**
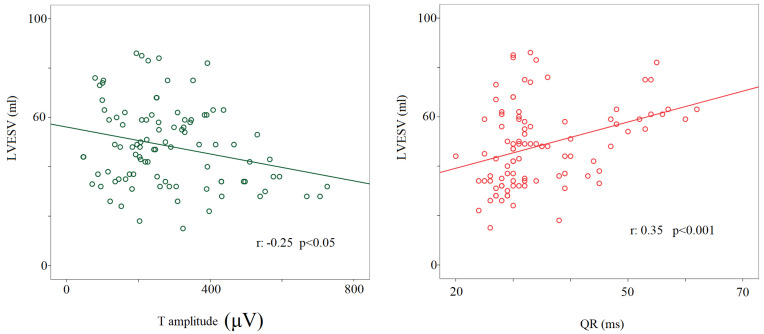
Correlation between left ventricular end-systolic volume or T wave amplitude (μV) (**left** panel) and QR interval (from the q to the R-wave) (ms) (**right** panel).

**Table 1 biology-12-00960-t001:** Power spectral analysis of P-P interval variability in different study phases.

	Rest	0° L-d	−45° Tilt	45° Tilt	90° Tilt	*p* Value
		Controlled Breathing				
**TP_PP_, ms^2^**	1166 (2217)	1246 (637)	1603 (3614)	1739 (1221)	1308 (1612)	ns
***p* value: TP_PP_ vs. TP_RR_**	0.05	0.05	0.05	0.05	0.001	
**VLF_PP_, ms^2^**	580 (845)	377 (637)	810 (2116)	784 (796)	780 (707)	ns
***p* value: VLF_PP_ vs. VLF_RR_**	0.05	ns	0.05	0.05	ns	
**LF_PP_, ms^2^**	251 (1252) @	285 (344) #@	366 (977)	437 (534)	487 (727)	<0.05
***p* value: LF_PP_ vs. LF_RR_**	0.05	ns	ns	ns	ns	
**LF_PP_ CF, Hz**	0.10 ± 0.02	0.10 ± 0.02	0.10 ± 0.02	0.09 ± 0.02	0.09 ± 0.02	ns
***p* value: LF_PP_ vs. LF_RR_CF**	ns	ns	ns	ns	ns	
**HF_PP,_ ms^2^**	167 (484) *@	219 (661) §	278 (32) ╪╫	164 (96) ●	100 (110)	<0.001
***p* value: HF_PP_ vs. HF_RR_**	<0.001	<0.05	<0.05	<0.05	<0.001	
**HF_PP_ CF, Hz**	0.30 ± 0.09 @*@@	0.25 ± 0.00	0.25 ± 0.06	0.25 ± 0.06	0.25 ± 0.09	<0.05
***p* value: HF_PP_ vs. HF_RR_ CF**	ns	ns	ns	0.001	ns	
**LF/HF_PP_,**	2.30 (2.37) @*@@	0.98 (1.14) @@§§	1.33 (1.27) ╪╫	2.57 (2.02) ●	4.43 (3.76)	<0.001
***p* value: LF/HF_PP_ vs. LF/HF_RR_**	0.001	0.05	0.001	0.05	0.05	
**LF_PP_, nu**	64 (25) @@	46 (27) @@§§	51 (29) ╪╪╪	64 (22) ●	71 (26)	<0.001
***p* value: LF_PP_ vs. LF_RR_ nu**	0.001	0.05	0.001	0.001	0.05	
**HF_PP_, nu**	31 (20) @@	45 (25) @@§	37 (19) ╪╫	24 (17) ●	16 (8)	<0.001
***p* value: HF_PP_ vs. HF_RR_ nu**	<0.05	<0.05	<0.05	<0.05	<0.001	

L-d: lay down position; PP: P-P interval; TP: total power; VLF: very low frequency; LF: low frequency; LF CF: LF central frequency; HF: high frequency; HF CF: HF central frequency; LF/HF: LF, HF ratio; nu: normalized units; ns: not statistically significant, *p* value > 0.05. *p* < 0.05 rest vs. 0° lie-down; @@ *p* < 0.001 rest vs. 0° lie-down; * *p* < 0.05 rest vs. −45° tilt; @ *p* < 0.05 rest vs. 45° tilt; @@ *p* < 0.001 rest vs. 45° tilt; @ *p* < 0.05 rest vs. 90° tilt; @@ *p* < 0.001 rest vs. 90° tilt; @ *p* < 0.05 0° lie-down vs. 45° tilt; @@ *p* < 0.001 0° lie-down vs. 45° tilt; # *p* < 0.05 0° lie-down vs. −45° tilt; § *p* < 0.05 0° lie-down vs. 90° tilt; §§ *p* < 0.001 0° lie-down vs. 90° tilt; ╪ *p* < 0.05 −45° tilt vs. 45° tilt; ╪╪ *p* < 0.001 −45° tilt vs. 45° tilt; ╫ *p* < 0.05 −45° tilt vs. 90° tilt; ● *p* < 0.05 45° tilt vs. 90° tilt.

**Table 2 biology-12-00960-t002:** ECG interval data in different study phases.

	Rest	0° L-d	−45° Tilt	45° Tilt	90° Tilt	*p* Value
		Controlled Breathing				
**RR, ms**	845 ± 114 **⸸⸸	840 ± 106 ##§§	912 ± 123 ╪╪╫╫	825 ± 130 ●●	697 ± 94	<0.001
**RR_SD_, ms**	34 (29)	35 (25)	40 (40)	41 (17)	36 (20)	ns
**PP, ms**	845 ± 114 **⸸⸸	841 ± 106 ##§§	917 ± 120 ╪╪╫╫	825 ± 130 ●●	697 ± 94	<0.001
**PP_SD_, ms**	34 (30)	35 (24)	40 (41)	42 (15)	37 (20)	ns
**P, ms**	123 ± 9 ⸸⸸	123 ± 12 #§	121 ± 11 ╫	121 ± 10 ●	114 ± 10	<0.001
**P_SD_, ms**	8 (2)	8 (2)	9 (4)	9 (3)	9 (2)	ns
**P_e_Q, ms**	55 ± 14 ⸸⸸	59 ± 15 §	59 ± 13 ╫	55 ± 14 ●	47 ± 14	<0.001
**P_e_Q_SD_, ms**	7 (3)	6 (3)	8 (3)	7 (4)	7 (3)	ns
**PQ, ms**	179 ± 17 ⸸⸸	186 ± 23 §§	182 ± 18 ╫╫	177 ± 18 ●●	161 ± 16	<0.001
**PQ_SD_,**	7 (3)	7 (3)	7 (4)	8 (4)	8 (3)	ns
**QR, ms**	34 ± 9	35 ± 9	35 ± 9	35 ± 10	35 ± 10	ns
**QR_SD_, ms**	3 (3)	4 (4)	4 (5)	4 (4)	4 (5)	ns
**QRS, ms**	66 ± 17	66 ± 17	67 ± 17	65 ± 19	64 ± 19	ns
**QRS_SD_, ms**	5 (4)	5 (4)	6 (5)	6 (4)	6 (5)	ns
**QT_p_, ms**	284 ± 20 *⸸	284 ± 18 #§§	298 ± 33 ╪╫╫	282 ± 22 ●	268 ± 23	<0.001
**QT_pSD_,**	6 (3) ⸋⸸	6 (4) ⸠§	7 (4)	8 (5)	8 (3)	<0.001
**QT_e_, ms**	370 ± 29 *⸸⸸	369 ± 32 #§	385 ± 26 ╪╫╫	369 ± 26 ●●	343 ± 32	<0.001
**QT_eSD_, ms**	7 (3) ⸡*⸋⸸	7 (2) #§	8 (3)	8 (4)	8 (5)	<0.001
**ST_p_, ms**	218 ± 29 ⸸	219 ± 26 #§	226 ± 32 ╫	217 ± 28 ●●	204 ± 28	<0.001
**ST_pSD_, ms**	6 (2) ⸸	6 (4) §	7 (3)	6 (3) ●	7 (2)	<0.05
**ST_e_, ms**	308 ± 33 *⸸⸸	308 ± 30 #§	318 ± 32 ╪╫╫	303 ± 32 ●●	284 ± 29	<0.001
**ST_eSD_, ms**	6 (3) *⸸	6 (3) #§	7 (3)	7 (3) ●	8 (3)	<0.05
**T_e_, ms**	87 ± 12	90 ± 12	92 ± 10	86 ± 10	86 ± 25	ns
**T_eSD_, ms**	7 (2) *⸋⸸	7 (2) #§	8 (2)	8 (3) ●	9 (3)	<0.001
**TeP, ms**	301 ± 86 *⸸⸸	293 ± 74 ##§§	362 ± 89 ╪╪╫╫	291 ± 97 ●●	196 ± 70	<0.001
**T_e_P_SD_,**	34 (29)	36 (23)	40 (39)	41 (13)	37 (17)	ns
**T, μVolt**	297 (348) ⸋⸸⸸	313 (233)⸠§§	298 (202) ╪╫╫	266 (196) ●	200 (127)	<0.001

L-d: lay down position; * *p* < 0.05 rest vs. −45° tilt; ** *p* < 0.05 rest vs. −45° tilt, +45° tilt and 90° tilt; ⸡ *p* < 0.05 rest vs. 45° tilt; ⸋⸸ *p* < 0.001 rest vs. 45° tilt; ⸡ *p* < 0.05 rest vs. 90° tilt; ⸋⸸ *p* < 0.001 rest vs. 90° tilt; ⸡ *p* < 0.05 rest vs. 0° lie-down; # *p* < 0.05 0° lie-down vs. −45° tilt; ## *p* < 0.001 0° lie-down vs. −45° tilt; § *p* < 0.05 0° lie-down vs. 90° tilt; §§ *p* < 0.001 0° lie-down vs. 90° tilt; ⸡ *p* < 0.05 0° lie-down vs. 45° tilt; ⸋⸸ *p* < 0.001 0° lie-down vs. 45° tilt; ╪ *p* < 0.05 −45° tilt vs. 45° tilt; ╪╪ *p* < 0.001 −45°tilt vs. 45° tilt; ╫ *p* < 0.05 −45° tilt vs. 90° tilt; ╫╫ *p* < 0.001 −45° tilt vs. 90° tilt; ● *p* < 0.05 45° tilt vs. 90° tilt; ●● *p* < 0.001 45° tilt vs. 90° tilt. ns: not statistically significant, *p* value > 0.05.

**Table 3 biology-12-00960-t003:** Multiple linear mixed regression analyses between beat-to-beat PP or RR intervals and other ECG Data.

	r	Slope	Intercept	*p* Value
	** *PP* **			
**P**	0.21	22.80	101.89	<0.001
**P_e_Q**	0.13	0.02	41.97	<0.001
**PQ**	0.24	0.04	143.78	<0.001
**QR**	0.08	0.01	39.76	<0.001
**QRS**	0.10	0.01	54.95	<0.001
**QT_p_**	0.64	0.11	191.04	<0.001
**QT_e_**	0.74	0.16	239.45	<0.001
**ST_p_**	0.48	0.10	136.08	<0.001
**ST_e_**	0.62	0.15	184.50	<0.001
**T_e_**	0.40	0.05	48.44	<0.001
**T_e_P**	0.97	0.80	−383.24	<0.001
	**r**	**Slope**	**Intercept**	***p* Value**
	** *RR* **			
**P**	0.24	0.03	99.15	<0.001
**P_e_Q**	0.12	0.02	41.86	<0.001
**PQ**	0.26	0.04	141.02	<0.001
**QR**	0.08	0-.01	38.81	<0.001
**QRS**	0.10	0.01	54.89	<0.001
**QT_p_**	0.65	0.11	190.47	<0.001
**QT_e_**	0.75	0.16	238.65	<0.001
**ST_p_**	0.48	0.10	135.59	<0.001
**ST_e_**	0.62	0.15	183.77	<0.001
**T_e_**	0.40	0.05	48.21	<0.001
**T_e_P**	0.96	0.80	−379.78	<0.001

R-R intervals (RR); P-P intervals (PP), from the beginning of two consecutive P waves; P wave intervals (P), from the start to the end of a single P wave; P-Q intervals (PQ), from the start of P to the Q waves; P-Q segments (P_e_Q), from the end of P to Q waves; Q-R intervals (QR), from Q to the peak of R waves; Q-R-S intervals (QRS), from Q to S waves; Q-T peak intervals (QT_p_), from Q to the peak of T waves; Q-T end (QT_e_), from Q to end of T waves; ST peak segments (ST_p_), from S to the peak of T waves; ST end segments (ST_e_), from S to the end of T waves; T peak T end intervals (T_e_), from peak to end of T waves; Tend-P intervals (T_e_P), from the end of T to start of P waves.

## Data Availability

The data underlying this article will be shared upon reasonable request to the corresponding author.

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
