# Peer review of "Effect of Head-Up/-Down Tilt on ECG Segments and Myocardial Temporal Dispersion in Healthy Subjects"

_biology, 2023, doi:10.3390/biology12070960_

Round 1

Reviewer 1 Report

There is a need for the manuscript to be thoroughly revised due to poor English grammar. I recommended that a native English speaker assists with improving the writing of this manuscript. The study protocol could be more easily explained with a figure that shows what took place, Improvement in the reporting of results in required. It is very difficult at times to follow what happened. In the Discussion, there is little mention of how the results of this study can be applied and where it helps to fill gaps in the existing literature. I also suggest adding information about future research studies that are required and the limitations of your study.

Further edits and comments are below.

Line 18: Please define ‘ANS’

Line 21: Do you ‘assuming’ rather than ‘desuming’?

Line 39: “Heart Rate Variability”

Lines 48-49: This part of the sentence should be a new sentence. It is also convoluted, please revise.

Line 58: “with respect to the link”

Line 68: “The aim of the present study, ….”

Line 104: Do you mean “T end-P intervals”?

Line 108: “…rest (lay-down)…”

Line 134: “tougher” – this seems like a poor word choice.

Line 141: “…with respect to the data..”

Line 184: Full stop after “rest”.

Line 203: “regardless of the study phase”

Line 231: “…sympatho-vagal imbalance…”

Line 241: Remove the closing bracket after “shortening”

Line 268: “Indeed the…”

Line 309: “…respect to the other study phases.”

Author Response

#Reviewer 1

We would like to thank the reviewer for his/her time spent for the revision. We have done our best to fulfill the request.

There is a need for the manuscript to be thoroughly revised due to poor English grammar. I recommended that a native English speaker assists with improving the writing of this manuscript. The study protocol could be more easily explained with a figure that shows what took place, Improvement in the reporting of results in required. It is very difficult at times to follow what happened. In the Discussion, there is little mention of how the results of this study can be applied and where it helps to fill gaps in the existing literature. I also suggest adding information about future research studies that are required and the limitations of your study.

 Reply: thank you for your useful remarks. We have modified the text and added a final, specific paragraph in order to explain the possible clinical application of our findings.

Further edits and comments are below.

 Line 18: Please define ‘ANS’

Reply: We have specified in the first line of the abstract

Line 21: Do you ‘assuming’ rather than ‘desuming’?

Reply: Yes, thank you.

Line 39: “Heart Rate Variability”

Reply: We have corrected

Lines 48-49: This part of the sentence should be a new sentence. It is also convoluted, please revise.

Reply: we have remodulated the sentence as follow “Noteworthy, each ECG segments (P-Q, P-R, S-T, and T-P) as well as ECG intervals (RR, PP, P wave, Q-R-S, Q-T) slightly oscillate due to the ANS control, which modulates sinus node activity, affecting ECG’s spectral components and coherences”

Line 58: “with respect to the link”

Reply: We have amended

Line 68: “The aim of the present study, ….”

Reply: ok

Line 104: Do you mean “T end-P intervals”?

Reply: T end-P interval

Line 108: “…rest (lay-down)…”

Reply: yes, amended

Line 134: “tougher” – this seems like a poor word choice.

Reply: the sentence was rewritten “considering the different study phases together all.”

Line 141: “…with respect to the data..”

Reply: ok, thank you.

Line 184: Full stop after “rest”.

Reply: thank you

Line 203: “regardless of the study phase”

Reply: ok

Line 231: “…sympatho-vagal imbalance…”

Reply: Right

Line 241: Remove the closing bracket after “shortening”

Reply: Ok

Line 268: “Indeed the…”

Reply: Corrected

Line 309: “…respect to the other study phases.”

Reviewer 2 Report

The authors submitted a research article in which they elucidated plausible relationship between the surface ECG and the autonomic nervous system modulation. With this aim, they included healthy subjects, was to describe deeply the physiological behavior of a number of ECG and noninvasive hemodynamic parameters in response to an ANS de/activation induced by different passive graded head-up/-down tests. They established that the ANS activity oscillations induced by postural de/activation of the cardiopulmonary reflexes during different passive graded head-up/-down tilt tests. The aim of the study is clear. The manuscript has a logical structure and well-organised subsections, which open up all aspects of the initial hypothesis. The conclusive part seem to be impressive. Although the findings are clear and concise, I would like to put some comments to dicuss.

1. The authors should give a brief comment at the end of the section "Discussion", whether this applied method of ECG evaluation improves maschine learning performances for continuous authomatic analysis of ECG, for instance for risk a sudden death assay, etc.

2. Clear practical recommendation should be added to the section "Conclusion".

Author Response

#Reviewer 2

We would like to thank the reviewer for his/her revision and suggestions. We have done our best to fulfill the requests.

The authors submitted a research article in which they elucidated plausible relationship between the surface ECG and the autonomic nervous system modulation. With this aim, they included healthy subjects, was to describe deeply the physiological behavior of a number of ECG and noninvasive hemodynamic parameters in response to an ANS de/activation induced by different passive graded head-up/-down tests. They established that the ANS activity oscillations induced by postural de/activation of the cardiopulmonary reflexes during different passive graded head-up/-down tilt tests. The aim of the study is clear. The manuscript has a logical structure and well-organised subsections, which open up all aspects of the initial hypothesis. The conclusive part seem to be impressive. Although the findings are clear and concise, I would like to put some comments to discuss.

  1. The authors should give a brief comment at the end of the section "Discussion", whether this applied method of ECG evaluation improves machine learning performances for continuous authomatic analysis of ECG, for instance for risk a sudden death assay, etc.

Reply: thank you for your suggestion! We have added this topic in a specific paragraph.

  1. Clear practical recommendation should be added to the section "Conclusion".

Reply: in the same paragraph we have added some suggestions about the clinical applications of our findings.

Reviewer 3 Report

The authors report the results of ECG analysis during tilt test with a patient population of 20 healthy subjects.

Novelty Compared to State of Research

Unfortunately, the authors do not provide a sufficient review of the state of research on the topic: The introduction only mentions 10 of the authors' own publications; results from other research groups are not reported. The referenced papers are not critically reviewed, particularly concerning disadvantages and open questions. Hence, the innovation of the proposed paper compared to the current state of research remains open.

What also is not addressed in the introduction is the motivation for the study. Just a collection of parameters that change during changes of body position does not establish a significant innovation. The authors should address how their findings can be used.

Methods

Although a large number of parameters is tested, the mulitple comparisons problem is not accounted for, e.g., by use of the Bonferroni correction. With the large number or tested parameters, chances are good that some of the group comparisons are significant just by chance.

Figures

Figures require captions below each figure (p. 3) and not with the caption on the next side (p. 4-5).

Figures should not have shadows.

For the caption-less figure on the bottom of p. 3, the source is not given if taken from another publication. Furthermore, there is no connection to the text; particularly the notation introduced in the text is not used.

The figure on p. 4 does not to serve any purpose. It is not even described or discussed and should therefore be deleted. The same applies to Fig. 1, which shows several well known ECG-patterns. The differences between them are neither described nor discussed, hence unused and to be deleted.

Tables

The tables are not very clearly arranged. In addition to a clearer layout, the authors should be consistent with omitting the zero before the comma and might consider numbered footnotes instead of the more or less conventional symbols used.

References

The largest drawback of the bibliography is that half of the references are self-citations [1-10, 12-13, 15-18, 31].

Furthermore, check your bibliography entries for consistent formatting, e.g., comma or no comma between year and volume, separation of authors [25], missing year in [35], abbreviated and not abbreviated journal name.

Further Formatting

Check the difference between hyphen and dash (minus sign).

Do not use headings on the last line of the page.

Check for double spaces, e.g., in the simple summary.

Check your references, e.g., l. 61, 94.

Author Response

#Reviewer 3

We would like to thank the reviewer for his/her revision and suggestions. We have done our best to fulfill the requests.

The authors report the results of ECG analysis during tilt test with a patient population of 20 healthy subjects.

Novelty Compared to State of Research

Unfortunately, the authors do not provide a sufficient review of the state of research on the topic: The introduction only mentions 10 of the authors' own publications; results from other research groups are not reported. The referenced papers are not critically reviewed, particularly concerning disadvantages and open questions. Hence, the innovation of the proposed paper compared to the current state of research remains open.

What also is not addressed in the introduction is the motivation for the study. Just a collection of parameters that change during changes of body position does not establish a significant innovation. The authors should address how their findings can be used.

Reply: We understand the reviewer's criticisms. Although there is a very old and not always pertinent literature on the subject, we were able to insert two further extremely pertinent recent bibliographic items. We have also added a specific sentence at the end of the introduction to better clarify the aims of the work and its clinical applicability

Methods

Although a large number of parameters is tested, the mulitple comparisons problem is not accounted for, e.g., by use of the Bonferroni correction. With the large number or tested parameters, chances are good that some of the group comparisons are significant just by chance.

Reply: The parameters evaluated were subjected to ANOVA for repeated measures. We have added a special sentence in the paragraph concerning the statistical analysis.

Figures

Figures require captions below each figure (p. 3) and not with the caption on the next side (p. 4-5).

Figures should not have shadows.

For the caption-less figure on the bottom of p. 3, the source is not given if taken from another publication. Furthermore, there is no connection to the text; particularly the notation introduced in the text is not used.

The figure on p. 4 does not to serve any purpose. It is not even described or discussed and should therefore be deleted. The same applies to Fig. 1, which shows several well known ECG-patterns. The differences between them are neither described nor discussed, hence unused and to be deleted.

Reply: We have eliminated the shadow, but, in our opinion, it seems clearer to keep the writings with the relative colors next to figure 1. Moreover, each figure already has a caption underneath. Figure 2 is made up of two distinct elements: the first is marked by the letter A and corresponds to an electrocardiographic tracing, with indications of the intervals and ECG segments which are analyzed in part B of the figure. The Numbers of figure A correspond to the relative intervals of figure B of which the spectral analysis was made. The figure is original and is not taken from any other work. In our opinion, the figures that the reviewer suggests removing from the text iconographically enrich the paper and make the method of evaluating the data treated in the text more understandable. We do not deem it appropriate to remove them.

Tables

The tables are not very clearly arranged. In addition to a clearer layout, the authors should be consistent with omitting the zero before the comma and might consider numbered footnotes instead of the more or less conventional symbols used.

Reply: We understand the reviewer's concerns about the symbols used, but the numbered notes could be misinterpreted as bibliographic references, so we prefer to keep this arrangement.

References

The largest drawback of the bibliography is that half of the references are self-citations [1-10, 12-13, 15-18, 31].

Furthermore, check your bibliography entries for consistent formatting, e.g., comma or no comma between year and volume, separation of authors [25], missing year in [35], abbreviated and not abbreviated journal name.

Reply: We understand what the reviewer said, but it must be said that this subject in particular, as already mentioned with regard to the introduction, has rarely been treated and by few authors in the recent period. The bibliography cannot be enriched extensively and we therefore base our evaluations on our old works.

Thank you very much for pointing out the typos in the references

Further Formatting

Check the difference between hyphen and dash (minus sign).

Do not use headings on the last line of the page.

Check for double spaces, e.g., in the simple summary.

Check your references, e.g., l. 61, 94.

Reviewer 4 Report

The novelty of the study and wide spectrum of performed analyses are very welcomed, considering that cardiopulmonary reflexes modualte autonomic balance. Thi That's why this paper deserves special attention. Major limitation of the study is small sample size, which is rather too low. I'm afraid that implemented methods are not adequate to phenotyping determination. Validity of the study findings is needed in larger studies which may confirm these findings. Another major concern about insufficient discussion. Authors collected wide set of parameters, but correlations between them are very poorly discussed.

Below few of my specific comments:

1. It is well known taht Physio Flow device provides poor data quality due to massive outlires and wrong calculations.

Could authors explain how they excluded outlires and recalulate analysed variables?

2. Authors need to expand on limitations.

3. 500 Hz ECG sampling is quite low for such kind of analyses, should be about 1 kHz. Could authors explaind that. 

4. How authors completed a power analysis, which algoryth was used within 5 minutes period?

5. introductions is lacking useful information and is underdeveloped. Doesn’t explain a background of the study.

6. Discussion is extremely underdeveloped.

Author Response

#Reviewer 4

We would like to thank the reviewer for his/her revision and suggestions. We have done our best to fulfill the requests.

The novelty of the study and wide spectrum of performed analyses are very welcomed, considering that cardiopulmonary reflexes modualte autonomic balance. Thi That's why this paper deserves special attention. Major limitation of the study is small sample size, which is rather too low. I'm afraid that implemented methods are not adequate to phenotyping determination. Validity of the study findings is needed in larger studies which may confirm these findings. Another major concern about insufficient discussion. Authors collected wide set of parameters, but correlations between them are very poorly discussed.

Below few of my specific comments:

  1. It is well known taht Physio Flow device provides poor data quality due to massive outlires and wrong calculations.

Could authors explain how they excluded outlires and recalulate analysed variables?

Reply: we have added the following sentence “Hemodynamic recordings have been double checked by echocardiographic tests and twelve-leads surface ECG; PhysioFlow system provided a self-correction to minimize artifacts and noises.” in order to better explain the method.

  1. Authors need to expand on limitations.

Reply: we have added a specific paragraph at the end

  1. 500 Hz ECG sampling is quite low for such kind of analyses, should be about 1 kHz. Could authors explaind that. 

Reply: As reported in the reference n. 12, Baumert et al Europace 2016 in their consensus clearly stated “A systematic comparison of sampling rates demonstrated that 500 Hz are sufficient while sampling rates of 200 Hz and below may artificially increase QTV values. Theoretical investigation of digitization noise and simulation studies also suggests that 500 Hz is a sufficient sampling rate for QTV measurement.”. Thus, we have always used this sampling method.

  1. How authors completed a power analysis, which algoryth was used within 5 minutes period?

Reply: Berger  formula was applied as reported in reference n. 19.

  1. introductions is lacking useful information and is underdeveloped. Doesn’t explain a background of the study.

Reply: we have better specified the aim and possible clinical application of the study.

  1. Discussion is extremely underdeveloped.

Reply: we have added another paragraph to the discussion. At the moment, the discussion is composed by a general paragraph and 5 different sub-paragraphs, which seem to be sufficient to the authors to elucidate all the results.

Round 2

Reviewer 1 Report

Well done on addressing my concerns and improving the quality of the manuscript.

Author Response

Thank you very much for your suggestion and appreciation.

Reviewer 4 Report

Accepted. Authors have adequatly responed to my remarks and suggestions.

Author Response

Thank you very much for your suggestions and comments.